# Sustainable Smart City—Opening a Black Box

**Mona Treude**

Energy, Transport, Climate Policy Division, Wuppertal Institute, Döppersberg 19, 42103 Wuppertal, Germany; mona.treude@wupperinst.org; Tel.: +49-202-2492-315

**Abstract:** Cities are becoming digital and are aiming to be sustainable. How they are combining the two is not always apparent from the outside. What we need is a look from inside. In recent years, cities have increasingly called themselves Smart City. This can mean different things, but generally includes a look towards new digital technologies and claim that a Smart City has various advantages for its citizens, roughly in line with the demands of sustainable development. A city can be seen as smart in a narrow sense, technology wise, sustainable or smart and sustainable. Current city rankings, which often evaluate and classify cities in terms of the target dimensions "smart" and "sustainable", certify that some cities are both. In its most established academic definitions, the Smart City also serves both to improve the quality of life of its citizens and to promote sustainable development. Some cities have obviously managed to combine the two. The question that arises is as follows: What are the underlying processes towards a sustainable Smart City and are cities really using smart tools to make themselves sustainable in the sense of the 2015 United Nations Sustainability Goal 11? This question is to be answered by a method that has not yet been applied in research on cities and smart cities: the innovation biography. Based on evolutionary economics, the innovation biography approaches the process towards a Smart City as an innovation process. It will highlight which actors are involved, how knowledge is shared among them, what form citizen participation processes take and whether the use of digital and smart services within a Smart City leads to a more sustainable city. Such a process-oriented method should show, among other things, to what extent and when sustainability-relevant motives play a role and which actors and citizens are involved in the process at all.

**Keywords:** smart city; sustainable smart city; sustainable urban development; innovation biography; sustainability; dissemination of knowledge; Sustainable Development Goals (SDGs)

## 1. Introduction

The creation of a Smart City is a highly complex process. Like Barlow said: "Smart cities will be laboratories for endless experiments. The experiments won't produce answers, but they will generate more questions. That's the nature of scientific process." [1] (p. 179). Cities today face many challenges, including the need to combine competitiveness and sustainable urban development. In addition, cities are considered to be particularly vulnerable to climate change, e.g., heat, heavy rain or pandemics [2]. Urban developers and city managers are increasingly trying to meet these challenges with smart solutions, like smart buildings that optimize their energy efficiency, intelligent traffic management or Open Data including eGovernment.

One of the weaknesses of the much-covered Smart City is the vagueness of a uniform definition. That makes it difficult to classify, especially for urban actors, but also for science. From the outside it is not clear whether the Smart City—to those bearing this title—is a vision, a marketing instrument, a political control instrument or something else entirely. The process towards the Smart City is often not easy to grasp, nor is the extent to which it creates sustainable urban development.

On the one hand, information and communication technologies (ICT) companies see the technological opportunities to offer new products (also) in urban areas. On the

other hand, cities are facing above-mentioned challenges that need to be solved and for which ICT companies (can) offer smart solutions. The challenge is to estimate whether the expected return or effect (e.g., resource conservation, better traffic flows, etc.) will make the (often large financial) effort worthwhile. The risk for such a (smart) solution always lies with the cities and their inhabitants, but the profits always benefit the ICT company, which is a major imbalance [3].

In principle, digitisation can help to reduce resource and energy consumption, but the opposite may also be true [4–6]. An intensive analysis of individual cities will show whether the process towards a Smart City was driven by motives of sustainable development and urban design. The assessment of this is not trivial, because in addition to ecological indicators there are numerous soft factors that influence sustainable urban development in the sense of Sustainable Development Goal (SDG) 11, for example in connection with social participation or political co-determination at the level of urban planning.

The next section provides a frame of reference that illustrates the complexity of Smart City and sustainable Smart City research and offers a definitional classification. Due to the numerous works on definitional clustering, this section can only show a part of it, but it outlines the underlying debate that has been going on for years. This is followed by the presentation of methodological approaches within Smart City and sustainable Smart City research, on which the method of the innovation biography is to be further built. This is followed by the presentation and classification of the method of innovation biographies and thus the proposal to use them as a new instrument of urban research and to reveal the process towards a sustainable Smart City. The aim is to open the "Black Box Sustainable Smart City" and chart a path towards a sustainable Smart City. In a next step, a selection of possible cities will be made which could be studied for such a method of analysis. The final step is a discussion of the method and a conclusion. The actual application to a city and the results will be part of a later article. If this relatively new method can be applied to a smart and sustainable city i.e., to the identification of actors, the dynamics of knowledge exchange and dissemination, relevant clusters and network connections, it can contribute to a better understanding of the processes behind a sustainable Smart City. As an explorative method, it is intended to complement the portfolio of urban research and Smart City research with regard to its procedural mechanisms.

## 2. A Brief Overview of Smart Sustainable City Definitions

In order to avoid repetition, the overview of existing definitions as well as methodological procedures and their findings (Section 3) draw strongly on two previously published reviews by Trindade et al. [7] and Ruhlandt [8]. These existing two reviews are used because they match the appropriate search criteria and have similar questions as those used here. They are supplemented and updated by scientific literature relevant to the questions at hand in the interdisciplinary databases Google Scholar, Science Direct and Scopus from the years 2018 to 2020. Trindade et al. [7] found 630 scientific articles from the years 2012–2017 using the search criteria "Smart City" and "Sustainability" in the databases Emerald Insight, Science Direct and Scopus and analysed 25 of these with regard to the above-mentioned search criteria. Ruhlandt [8] lists ten years (in the period from January 1997 to May 2017) of research on Smart City in connection with governance using a transparent search protocol in the databases Business Source Complete (EBSCO), Web of Science and ABI Inform Global [8] (p. 2).

In all this research on Smart City from 1997 to 2020, it is remarkable how close the Smart City definitions and descriptions are to those of a sustainable city, even before the publication of the SDGs in 2015. However, after 2015, SDGs only play an indirect role in the definition of sustainable Smart Cities and are not mentioned directly. Lytras and Visvizi, speak in this context of the framing sustainability imperative [9] (p. 5). Vanolo [10] already defines the Smart City in terms of SDG 11, as a city that is not only efficient and technologically advanced, but also green and socially inclusive [10]. A few years earlier in 2011 Caragliu et al. [11] described the Smart City as an instrument for socio-ecological

transformation within cities and defined a city as "smart" if it manages to use the technical means to conserve finite resources, which is also very reminiscent of the definition of a sustainable city in terms of SDG 11 [11]. Often, when a Smart City is first defined or described, "sustainable", in the sense of an ecologically responsible city, appears even before "progressive", or "economically productive", or as a tool for sustainable development (see inter alia Batagan, 2011 [12], Nam and Pardo, 2011 [13], Yigitcanlar, 2014 [14], Lee et al., 2014 [15], Yigitcanlar, 2015 [3], and after 2015 among others Meijer and Bolívar, 2016 [16], Lara, Moreira Da Costa, et al., 2016 [17], Ibrahim et al., 2016 [18], Fernandez-Anez et al., 2018 [19], or in response to the challenges of a city Yigitcanlar, 2018 [20]). According to de Jong et al., the indicated city category also determines the underlying orientation with regard to the design towards a sustainable or Smart City. They warn of competitive pressure among cities leading to technical problem-solving under the label "Smart City" without impacts on quality of life and sustainability becoming clear [21] (p. 36). Similarly, de Jong et al. [21], Lara, Moreira Da Costa, et al. [17] emphasize the relevance of the choice of the terms "smart", "sustainable", "intelligent", "green", "learning" or "zero-carbon" city, as it reflects the focus and operational emphasis [17] (pp. 6–7).

Newer definitions of the sustainable Smart City have been proposed several times since 2015, including in Ahvenniemi et al. [22], Aina, [23], Bibri and Krogstie [24], Yigitcanlar, Kamruzzaman, et al. [25], Yigitcanlar, Han, et al. [26] and especially Martin et al. [27], who analysed the tension between a Smart City and a Sustainable City and identified a key element: "A key finding is that the potential to empower and include citizens represents the key to unlocking forms of smart-sustainable urban development that emphasise environmental protection and social equity, rather than merely reinforcing neoliberal forms of urban development." [27] (p. 269). Literature reviews on the topic of Smart City repeatedly refer back to the same researchers and their definitions of the Smart City: Giffinger et al. [28], Yigitcanlar and Lee [29], Vanolo [10], Gil-Garcia et al. [30], Meijer and Bolívar [16], Meijer et al. [31] etc. In 2016 Dameri and Benevolo, still describe the smart city as an unfinished practice [32] (p. 693).

Lara, Da Costa, et al. [17] propose a definition according to which a Smart City is a "[...] community that systematically promotes the overall wellbeing for all of its members, and flexible enough to proactively and sustainably become an increasingly better place to live, work and play." [17] (p. 9). This echoes the approach to urban design of urban planner Jan Gehl [33], (initially) leaving the technological element aside. This frees cities from the competitive pressure mentioned above being forced to become ever more digital. However, the term smart almost always has a connotation of technology. Indeed, one could wish for a definition that focuses on people, their quality of life and sustainable development along the SDGs, but what makes it a Smart City? According to Söderström et al. [34], the Smart City is a trademark registered on November 4, 2011 by IBM. They trace the path of IBM's Smarter Planet campaign and thus the commercial origin of the Smart City idea [34] (p. 307). In this sense, the concept of "Smart City" is a commercial offer made to cities by a large ICT company (IBM) and thus a technology push. In a combination of qualitative, quantitative and desktop-based research, Lytras et al. [35] investigated, among other things, which urban smart services are used and desired by citizens at all. Marsal-Llacuna and Segal [36], even criticize the (previous) Smart City projects on the market as being over-technologized and without the added value for the citizens they promise [36] (p. 129). Haarstad [37], found similar tendencies in an investigation of EU-funded Smart City projects. The programmes and projects are "[...] driven more by concerns for economic growth and innovation than by environmental sustainability per se." [37] (p. 434). "Sustainability does not appear to be a very important motivating driver. Yet the 'sustainability component' of the Smart City agenda becomes more apparent the closer we come to the city level" [37] (p. 435). Yigitcanlar, Han, et al. [26] describe our current knowledge about Smart Cities as limited and write about speculative and unrealistic expectations [26] (p. 8). Following the lack of conceptualisation, we measure the term

smart too much in terms of "[ . . . ] technological smartness rather than human/decision smartness." [26] (p.8).

The various definitions above illustrate the complex challenge of finding a definition that is holistic, people-focused and oriented towards sustainability while including the technological aspects in a fitting form. A correspondingly large amount of scientific literature is devoted to precisely this challenge. As the definition and description of the Smart City shapes the underlying vision and strategy this is relevant for practice as well. Current research increasingly focuses on the question of whether the Smart City contributes to more resource conservation i.e., leads to more sustainable development, which does not seem to be the case so far: "The findings revealed that the current smart city efforts are not adequate enough to combat the challenges of the Anthropocene. Smart city policy, planning and development practice, at its best, is a zero-sum game for sustainability." [38] (p. 107). In some cases, rebound effects may even lead to impacts, e.g., requirements when the use of technology drives energy demand [22] (pp. 241–242). The method presented in Section 3.2 will help to find out whether aspects of sustainability are considered at all in the development process towards a Smart City, whether sustainability-relevant actors are involved, and whether the process towards the Smart City is guided by the idea of sustainable development. As a bias we use the following definition of a Smart City, drawing on the existing proposals: "A city is a Smart City if it uses (smart) technologies to better meet the challenges of the 21st century. This includes in particular sustainable urban development in the sense of SDG 11".

## 3. Methods Employed in Smart City Research and Smart Sustainable City Research and Some of Their Findings

Ruhlandt's [8] analysis shows various methodological approaches [8] (p. 11). He lists the articles examined in four categories. Methodological approaches that are relevant to the present questions can be found in the components of Smart City governance, more specifically in the sub-category "Structures and Organisations, Process and Roles and Responsibilities" [8] (p. 4). After reviewing and clustering the articles, Ruhlandt [8] comes to the conclusion that the identified components in the scientific literature mostly correspond to theoretical derivations and that the empirical evidence is insufficient or even completely lacking [8] (p. 8). Nevertheless, there are key points that need to be included when analysing the innovation process towards a Smart City both in theory and methodology: Taylor Buck and While [39], for example, describe the Smart City as an innovation among the overlapping interests of science, administration and the ICT companies, focusing on actors and innovations [39]. Dameri and Benevolo [32] examine the actor linkages within 117 Italian cities and come to the conclusion that to date there are no good examples of participation mechanisms in Smart City Governance [32] (p. 704). Gil-Garcia et al. [30], on the other hand, emphasize cross-sector networks and actor cooperation and Dameri and Ricciardi [40] found out in a multi-method mix that without intelligent coordination of the individual subsystems within the Smart City, generated knowledge does not serve as a catalyst for the Smart City, but is simply lost because it is too difficult to manage [40] (p. 877).

Five articles from Trindade et al. [7] are of particular interest for this article because they contain framework conditions for using digitalisation as a driver for sustainable development in cities like Angelidou, [41]. Based on innovation economics theory, there is also a focus on the process of a Smart City, for example in Lee et al. [15], or on elaborating the relevance of the cooperation, like in Marsal-Llacuna and Segal [36], Bayulken and Huisingh [42], Lara, Moreira Da Costa, et al. [17]. Lee et al. [15] who explore the relevance of urban networks: "The study also reveals that diversifying complementary networks and devices also help to accelerate adoption; in this respect, the city's own network capacity and usage are critical elements for both cities." [15] (p. 98). Although it is not surprising that the more people are involved in urban development processes, the greater the acceptance of the programmes and projects mentioned; these forms of participation are neither a matter of course nor a given. In their study, Marsal-Llacuna and Segal [36] state that citizens are

the key to the success of a Smart City (in the sense of a Smart Sustainable City) and that a collaborative approach is necessary for Smart City development [36] (p. 131). They develop an ICT-supported multi-stakeholder collaboration approach beginning earlier and going deeper than the classic dialogue processes with stakeholders [36] (p. 131).

There is a relatively large amount of theoretical and meta-level research (document and literature analysis), including research on sustainable Smart Cities, as exemplified by Neirotti et al. [43], where research articles, reports and published studies on Smart Cities, sustainable cities, sustainable urban development, sustainable Smart Cities and urban development theories are analysed. The concrete process towards a sustainable Smart City remains vague: the vision, the stocktaking, strategy and implementation, monitoring and evaluation, who is involved, who shares knowledge with whom, and which actors work together and how. As detailed above, there are many good theoretical considerations, but few empirical studies, especially regarding the creation process of a Smart City. Some of the works mentioned above are intended to show the directions Smart City research has taken so far, but are of course only extracts from the research and should not be considered as complete. One potential method, which has not yet been used in (sustainable) Smart City Research, is the innovation biography. It will be explained in more detail below.

### 3.1. Method: Innovation Biography

In order to find out how a Smart City process is designed and whether and how it contributes to a more sustainable urban design, a method should be used that can capture this process. The method presented here for such a process-oriented analysis is the innovation biography, which can provide qualitative ex-post analyses of innovation processes, in this case the spread of regional knowledge dynamics, and the multi-level connections of the actors (ICT companies, city administrations, urban planners, citizens, research institutions) over the course of the creation and implementation of a Smart City. It shall identify the relevant networks and clusters necessary to make the city not only smart but also more sustainable through the use of smart technologies. Beyond the direct involvement of citizens in the planning processes, others such as administrators from various departments, environmental NGOs or scientific institutes might have been addressed to support the development process towards a *sustainable* Smart City.

Since the 1960s at the latest, innovation research, building on the work of Joseph Schumpeter, among others, has gained far-reaching insights into many research disciplines. However, there is much more to be taken from innovation research besides the definition of the Smart City as an innovation e.g., how the process towards a sustainable Smart City is designed. Within the different phases of an innovation—discovery, invention, development and dissemination—numerous innovation processes can be identified that are relevant to the innovation sustainable Smart City, such as the recompositing of teams, the formation of clusters, the dissemination and sharing of knowledge and applications, etc. [44] Other researchers have already described the Smart City as an innovation, like Angelidou, [41]. According to Lara, Moreira Da Costa, et al. [17], innovation is "[...] an integral part of the concept of smart cities." [17] (p. 3). Besides technological possibilities, the sustainable Smart City is a new combination of existing or new ideas, skills, abilities, and also resources and can therefore be called innovation.

Every Smart City is embedded in a region and can therefore be understood as a regional innovation system i.e., an innovation that has emerged from the participation of various institutions, companies, research facilities and their cooperation with each other. These systems have a strong regional component and the exchange of knowledge and the density of cooperation is usually more intensive the closer their locations are to each other [45]. For the analysis of a sustainable Smart City, it is important to take this system perspective, because a system as network connects actors and activities, such as the city employees and the ICT companies that offer their smart solutions. These connections of the system can be used productively, or path dependencies can even block the whole process and a so-called "lock in" effect is created, which in the worst case can bring the whole

project (towards a sustainable Smart City) to a standstill. An important feature of any innovation is the recognition that action and interaction are driving factors for successful innovation, both in existing networks and clusters and with the environment in which the innovation is created [44]. Following these principles and analysing the sustainable Smart City as an innovation, the innovation biography can be employed as a valuable tool. It was developed in the early 20th century as a form of biographical research in sociology (see, among others, Kohli [46], Kneer and Schroer [47] (p. 85f)) and more recently by the Institute for Work and Technology (IAT) in different contexts. This method provides the possibility to outline the process of an innovation, to make precisely those networks and structures visible that are essential for an innovation and to identify actors and how the knowledge needed for the innovation is acquired, shared and diffused. This method, which has not (yet) been applied in Smart City research, is intended to generate an insight into the innovation process, i.e., the process of creating a sustainable Smart City, and thus open the "Black Box Sustainable Smart City".

Many cities call themselves Smart City, and expect this to make them more sustainable, without it being possible to see why from the outside. Certainly, they use smart technologies. However, to find out which ones are used for what purpose and out of which motivation, a process-oriented method is required. The biography will show which ideas, held by which of the responsible actors, are behind the sustainable Smart City and who has participated in the process. It will also reveal at what point in time additional actors and companies or even research institutions were integrated and how the generation of knowledge and knowledge diffusion took place. Actors and knowledge pertaining to sustainability will be of particular interest in the analysis. To this end, innovation biographies reconstruct the narrative of an innovation process from conception to implementation. For this they analyse, among other things, the territorial knowledge dynamics over time, space and individuals [48]. As a research tool, innovation biographies analyse these processes and dynamics from a micro-level perspective, and in doing so, they capture the "social relations, contextual settings, and the cross-sectoral and multi-local reach of knowledge developed and applied in innovation processes" [48] (p. 220).

Data collection for the development of an innovation biography involves a multi-step process. First, in the preparatory phase, a case study will be selected (see Section 4) and qualitative desktop research will be conducted on it in order to obtain freely accessible material on the innovation process. Then, an in-depth narrative interview with (a) key initiator(s) of the innovation is performed, in order to complement the narrative. As Butzin and Widmaier [48] note, this interview is the backbone of the innovation biography, as it is the "*essential instrument in operationalizing the open and explorative approach of innovation biographies*" (p. 225). From here, the social network of the innovation is explored through subsequent desktop research and additional interviews. The additional research is based on the egocentric network analysis, in which one point—in this case the innovation sustainable Smart City—is evaluated and described via its relationship to organizations and persons. Finally, all obtained information regarding the innovation is combined and analysed comprehensively. This allows for the creation of an innovation biography with special attention paid to the spatial and temporal dimensions of the innovation's development. These dimensions can then be visualized, as inspired by Butzin and Widmaier [48]. Here, geographical and time data can be displayed in combination. This allows the network to be derived and the actors involved to be displayed at specific points in time within the process.

The method of innovation biographies follows both a deductive and an inductive research approach. Through the open interview situations, new insights can be generated on the one hand (inductive approach) and at the same time previously made assumptions can be checked, which follows a deductive approach [48]. The creation of an innovation biography includes desktop research, narrative interviews as well as semi-structured interviews, an egocentric network analysis and is therefore a triangulation of mixed methods. The focus is on the narrative and egocentric interview regarding the process of innovation.

The creation of such an innovation biography takes place in eight steps and is shown in more detail in Figure 1 and Table 1 [49,50]:

1. Selection of the case study (see Section 4 for possible case studies).
2. Making an appointment for the first narrative interview.
3. Document analysis of the case study to capture the context of the innovation (homepage, freely accessible articles, planning documents, etc.).
4. Conduct the first narrative interview, in the best case with the person who was in charge of the innovation process.
5. Create a first egocentric perspective of the life story and, as a result of the drawing of a so-called egonet, investigate further actors and discussion partners.
6. Further interviews with central actors of the innovation with semi-structured guidelines, based on questions that arose from the first narrative interview of step 4. Here, further perspectives of the innovation process are collected in order to complete new information through the innovation process. This creates a second, third, etc. version of the innovation.
7. Supplementary interviews, for individual questions that have arisen from the previous interviews. It has been shown that up to 15 interviews may be necessary to complete the innovation biography.
8. In the last step, the reconstruction of the innovation biography begins by evaluating and presenting all the data previously collected from the document analyses and the interviews.

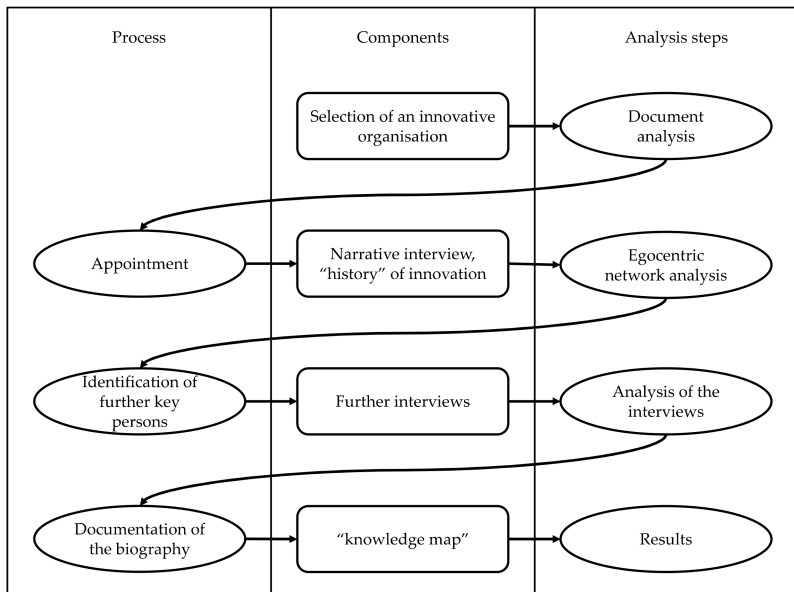

**Figure 1.** Research process within an innovation biography. Source: Own illustration according to Butzin, (p. 194), in Dannenberg, Peter (Ed.) et al. [50].

The selection of different interview and discussion partners has a corrective function, as they may be tempted to gloss over any challenges and obstacles that may have arisen or to euphemistically present one's own role [50].

The various steps are structured as follows [48]:

1. An individual level, where the key actor(s) can express their views.
2. Data on a structural level, i.e., the actors involved, modes, frequency and geographical distribution of interaction, are obtained by egocentric network analysis and construction of the time-space path (see Figure 2).
3. Document analysis as a third component has the function of enriching the biography by understanding sectoral characteristics and the contextual level.

Subsequently, these data from the different surveys of egocentric network analysis, geographical analysis and the preceding desktop research are triangulated to a biography [48] (p. 226). The different interviews form the individual level of the innovation biography.

Up to now, such a biographical approach has not been employed in Smart City research. Although there are also research studies that deal with the process of a Smart City, both theoretical and practical analytical approaches, the attempt to sketch the process using case studies is a new approach that can be derived from a suitable example (see Section 4) and can help to understand the process towards a sustainable Smart City in practice and, in the best case, to steer it effectively towards sustainability. The sustainable Smart City innovation biography should help to show the different types of knowledge relevant for an innovation, which also include the spontaneous search and sharing of knowledge. "On the whole, the knowledge and innovation economy is an essential driver of the smart city discourse" [41] (p. 99).

### 3.2. Method: Innovation Biography. Origin and Previous Areas of Application

The empirical recording of innovation dynamics is a central component of evolutionary economics [51]. Evolutionary economics does not refer to static conditions, but to dynamic processes, which also include knowledge creation as a process of searching for new knowledge and its diffusion [49]. Innovations are accordingly "[...] regarded as the output of an innovation production function whose most important input is new technological knowledge." [51] (p. 3). During the innovation process this is enriched by further knowledge, so-called experience knowledge, which is created through trial and error [51] (p. 3 ff.).

Innovation biographies are (up to now) used to record innovations over a period of time for companies and organisations, looking at both knowledge-based skills and functioning interactions of the actors involved (e.g., for renewable energies [52], in the construction industry [53], in nanotechnology [48]). They represent a qualitative research approach from the field of social research, which enables a holistic and detailed investigation of the territoriality of knowledge dynamics. The idea of innovation is investigated over time by analysing the interactions of the innovation actors and using the acquired data as an inductive approach. The special feature of this method is the possibility to map the entire innovation process, from the first ideas to the implementation. That dynamic approach focuses on the generation, processing and use of knowledge [49].

### 3.3. Method: Innovation Biography. Transfer to Urban Research

When and how an innovation comes to the market depends on various factors. If we approach the concept of a sustainable Smart City through the innovation theories of Schumpeter from around 1930, as well as Schmookler from 1966, as Angelidou [41] does, we share the following conclusion: Angelidou [41] postulates the need to invest not only in digital infrastructure but in knowledge diffusion and capacity building to enable citizens to partake in the innovation economy. In addition, Angelidou [41] warns against the current dominance of supply-oriented Smart City solutions, as these approach the problems of the city in a fragmented way and thus detach them from their social contexts. Innovation biographies can provide the required insights into the processes behind the Smart City and show whether the successful sustainable Smart City selected in the case study is more supply-oriented or demand-oriented, as demanded by Visvizi and Lytras [54]. Here, the diffusion theory of innovation by Rogers [55] could also provide interesting additional insights.

In order to write an innovation biography for a sustainable Smart City process and then, in the best case, to present it visually, the starting point of the process must first be determined (see Figure 2). Then a key person is identified who initiated the Smart City process or at least accompanied it from the beginning. This could possibly be a position created specifically for this purpose, such as that of a Smart City manager, or could be based with the mayor. The identified key person describes the Smart City innovation

process from their perspective with the help of a narrative interview. From the narrative of the first key Figure, other actors and networks involved in the process must then be identified and also interviewed. These groups of people may be participating citizens or ICT companies, possibly also other departments within the city. They can be identified from the first interview. This can also be continued in the sense of a snowball system and entail further interviews until the process towards a sustainable Smart City is comprehensible. Depending on the quality of the data obtained from the first narrative interview, either narrative interviews are then conducted again, or the interviews are continued in a semi-structured manner. The aim is to use the further interviews to supplement the information from the first interview and to complete the biographical image of the sustainable Smart City as a process. Beyond the interviews, research and planning documents are included in the analysis of the process. Thus, the innovation biography is drawn around the egocentric initial interview. In the further course of the process, the data obtained from the different levels are triangulated to form a biography and the history of the sustainable Smart City's creation is 'told' [56] (p. 40–42). Based on this, the social relationships can be analysed as well as the driving and inhibiting factors for the development of the sustainable Smart City, or critical events or contextual settings. Furthermore, the innovation biography of the sustainable Smart City will show what motivations led to the creation of the Smart City and the role sustainability played in these considerations. Figure 2 shows an exemplary visual representation for a sustainable Smart City process.

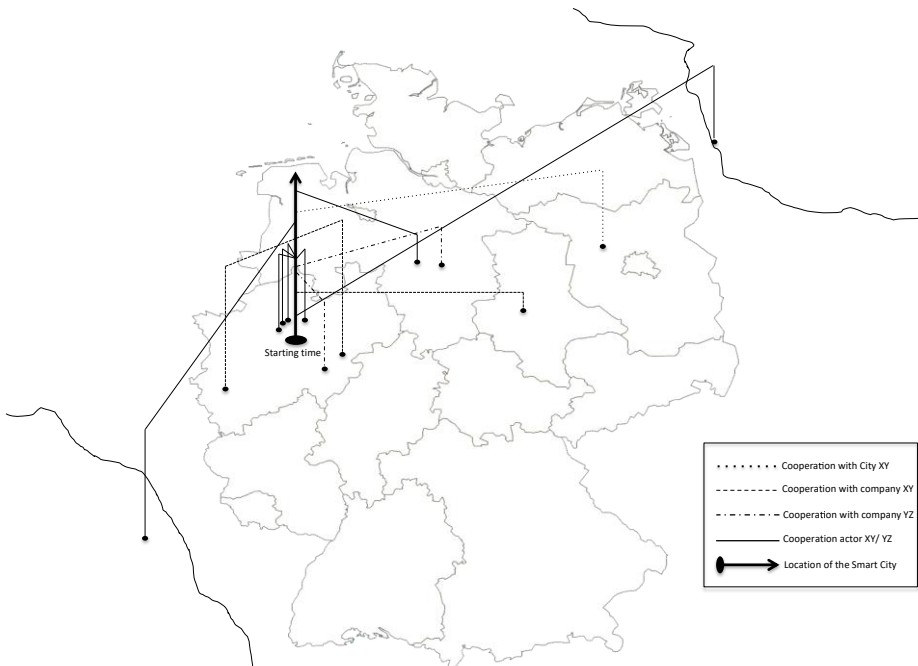

**Figure 2.** Exemplary visual representation of a sustainable Smart City process. Source: Own illustration according to Butzin [56] (p. 46).

The innovation biography will show how the sustainable Smart City is created in its network and its processes. It will show who is involved in the process and which projects support others. Whether citizens are involved at all, and if so at what point in time, in the process of creating the Smart City is also an important question from a sustainability perspective. According to Dameri [57], the process of digitisation of a city does not arise from movements of citizens towards the economic, ecological, social and urban development problems within the city. It seems rather common that citizens are the last to learn about their digital city as a Smart City. Rather, the Smart City is one of many waves of technological digitalisation that are coming over cities in the form of projects, initiatives, funding programmes and high-tech companies, according to Dameri [57]. A

biographical depiction of the process can show the extent to which the sustainable Smart City is also designed bottom-up. The challenges in transferring the innovation biographies method are outlined in Table 1 below, along with the dedicated procedures and how to deal with the potential challenges for transfer.

**Table 1.** Procedure and challenges of creating an innovation biography of a sustainable Smart City.

| Innovation Biographies | | | |
| --- | --- | --- | --- |
| **Procedure and Working Steps** | **Conditions for the Transfer to Smart City Research** | **Potential Challenges** | **Dealing with Challenges** |
| Case study selection | Identifying a successful sustainable Smart City | For the definition "successful", grey literature is used: Smart City Rankings | Comparison of ranking systems with regard to indicators, inclusion of desktop research |
| Document analysis for the case study | Identification and collection of all relevant documentation | Planning documents and municipal decisions may not all be freely accessible | Apply for relevant documents in the city |
| Selection of a key actor | Often long periods towards a sustainable Smart City, there may be several key players | Conducting several interviews already in the first round | Comparison of the processes described in the interviews |
| Conducting a narrative interview | No special conditions for the transfer to Smart City Research | Elaborating the process towards the sustainable Smart City from the experiences of relevant actors | Strengthening narrative demand phase and evaluation on the basis of narrative-structural methods |
| Egocentric network analysis | The hub of the network analysis is the sustainable Smart City itself | The egocentric network analysis is selective with regard to the actors and networks involved | Combination of the ego network with geographical and temporal data |
| Further interviews | No special conditions for the transfer to Smart City Research | Identifying all relevant actors from the first interview | Strengthening narrative demand phase from step 4 and deriving further actors from the following interviews |
| Triangulation of the data | Collect all data relevant to the innovation biography from and with involved actors | Aggregation and analysis of different types and amounts of data | Using triangulation as an approach to link the different research perspectives |
| Creation of the innovation biography | No special conditions for the transfer to Smart City Research | Identification of a start and an end point for the innovation | Triangulation of the methods |
| Analysis of procedural factors | No special conditions for the transfer to Smart City Research | Derivation of procedural factors only determinable for the analysed city | Transferring of the analysed factors and cross-check in other sustainable Smart Cities or continuation of the innovation biographies in comparable sustainable Smart Cities |

Source: Own representation according to Butzin 2014 [56].

In Conclusion there has been research in the field of sustainable Smart City to date that describes precisely these interactions and the cooperation of the actors as a relevant and driving factor, for example Batagan [12], Nam and Pardo [13], Ibrahim et al. [58] or Fernandez-Anez et al. [19], but on a meta-level, without analysing the dissemination of knowledge among the actors and the cooperation more closely. The identified research gap shall be closed with the help of a method from biographical research, the innovation biography. To further describe this research gap: "A multilevel analysis of the processes concerned, however, remains necessary, because many firms have relations with firms in other parts of the world." [59] (p. 1033). "The continuous renewal of the knowledge base

requires regionally co-operative structures and networks in which governments, scientific organisations and firms participate" [59] (p. 1033).

## 4. Case Study Selection

The selection of one or more suitable case studies is part of the preparatory phase of an innovation biography (see Table 1). In order to find out how the process towards a sustainable Smart City is designed, the selection of a suitable city is very important. It must meet the criteria of a Smart City and the criteria of a sustainable one. In order to be able to identify suitable case studies, well-known city rankings are used, which compare cities with each other through different indicators (see Table 2). For the selection of the case study, Table 2 lists and compares well-known city rankings, on smart cities and sustainable cities. This selection was supplemented by the "Quality of Living City Index", as the discussion on the sustainable Smart City discussion shown in Section 2 has shown that quality of life is an important target dimension. Table 2 also shows, for example, that the underlying indicators of the City in Motion Index (CIMI) and the Smart City Strategy Index are very similar to the Sustainable Cities Index (SIC). This would fit the impressions from chapter 2 that current smart city strategies go beyond the technology-centred view and try to combine smart and sustainable. The cities to be selected as case studies should therefore be placed high in the Smart City Rankings as well as in the Sustainability City Rankings, as these cities have (probably) managed the balancing act between smart and sustainable.

**Table 2.** Smart-, Sustainable- and Liveability City Rankings and their underlying indicators.

| | Index 2019 | Index 2018 | Index 2019 | Index 2019 | Index 2019 |
|---|---|---|---|---|---|
| City Ranking Places | City in Motion Index (CIMI) [1] | Sustainable Cities Index (SIC) [2] | Quality of Living City Index [3] | Global Liveability Index [4] | Smart City Strategy Index [5] |
| 1 | London | London | Vienna | Vienna | Vienna |
| 2 | New York | Stockholm | Zurich | Melbourne | London |
| 3 | Amsterdam | Edinburgh | Munich | Osaka | St. Albert |
| 4 | Paris | Singapore | Auckland | Calgery | Singapore |
| 5 | Reykjavik | Vienna | Vancouver | Sydney | Chicago |
| 6 | Tokyo | Zürich | Düsseldorf | Vancover | Shanghai |
| 7 | Singapore | München | Frankfurt | Toronto | Birmingham |
| 8 | Copenhagen | Oslo | Geneva | Tokyo | Chongqing |
| 9 | Berlin | Hong Kong | Copenhagen | Copenhagen | Shenzhen |
| 10 | Vienna | Frankfurt | Basel | Adelaide | Paris |
| Index Indicators | Human capital, Social cohesion, Economy, Public management, Governance, Environment, Mobility and transportation, Urban planning, International outreach, Technology | People (Social), Planet (Environmental, Profit (Economic) | Recreation, Housing, Economic environment, Consumer goods availability, Public service and transport, Political and social environment, Natural environment, Socio-Cultural environment, School and education, Medical and health considerations | Stability, Healthcare, Culture and Environment, Education, Infrastructure | Budget, Buildings, Energy and Environment, Mobility, Education, Health, Government, Infrastructure, Policy and Legal Framework, Stakeholders, Coordination, Plan |
| Goal dimension | Smart | Sustainability | Liveability | Liveability | Smart |

Source: Own representation. [1] https://media.iese.edu/research/pdfs/ST-0509-E.pdf; [2] https://www.arcadis.com/de/germany/unsere-perspektiven/sustainable-cities-index-2018/germany/; [3] https://mobilityexchange.mercer.com/Insights/quality-of-living-rankings; [4] https://www.eiu.com/topic/liveability; [5] https://www.rolandberger.com/de/Publications/Smart-City-Strategy-Index-Wien-und-London-weltweitfortschrittlichste-Städte.html.

The innovation biography will show to what extent these cities used sustainable development as an orientation for their smart city process. The process perspective will show at what point, for example, climate protection managers were involved, or the expertise of sustainability scientists was included, etc.

Vienna and London are two of the cities that are in the top positions in the rankings with both the target dimension smart and the target dimension sustainability (as of 2018/2019, see Table 2), so they are very well suited as case studies. In addition, Vienna has been considered the most liveable city for several years now (see Quality of living City Ranking by Mercer LLC). For the rankings, various indicators describing the city such as wealth, environment or infrastructure are combined into an index [60]. However, urban development is not status or a timeline, but a much deeper process hard to capture with the available data. In order to analyse it, a suitable process analysis is needed. Through the process perspective, conducting and analysing innovation biographies offers a deep insight into knowledge creation, as well as spatial structures and processes in innovation.

Each city has its own needs and challenges (city density, topography, infrastructure, etc.). However, the mechanisms for appropriate networks and clusters are transferable, as is who contributes what knowledge and how it must be controlled and stored so that everyone has rights and access to this knowledge. Other drivers and success factors have to be identified as well, including the design of citizen participation or the appropriate use of ICT companies. Interdependencies and iterative loops within the development towards a sustainable Smart City can also be illustrated in this way, as well as possible regressions and learning experiences drawn from them. With the strong actor perspective, causalities in the innovation emergence can be shown and analysed [56] (p. 48).

## 5. Discussion

Every city is a constantly evolving entity that attempts to meet the challenges of the 21st century in the most diverse constellations of actors. This complexity is well-served by the method of innovation biographies when trying to understand the processes that make smart cities successful and sustainable. The use of various standardised methodological tools, such as biographical narrative, interviews, an egocentric network analysis and triangulation, enables comparative research and the comparison of results. Finally, every qualitative research design has to meet the usual quality criteria. Thus, the reliability of the results and their generalizability are often doubted. This is to be discussed and ensured by the criterion of intersubjective traceability in addition to other criteria for ensuring quality in qualitative social research. Thus, the entire research process, including data collection and data evaluation, must be designed to be comprehensible and verifiable [61] (pp. 231–233).

Every research design has limitations. This also applies to the innovation biographies: In order to create an innovation biography, it is necessary to determine a starting point i.e., the beginning of the sustainable Smart City process. This may not be easy to find out, as it could be multiple processes or multidimensional processes. The same applies to the end of the process. Moreover, although the approach is very detailed, it is not representative. The results cannot easily be transferred to other cities, but must be adapted as process components and into clusters. [56] (p. 51).

In addition, the quality of the information for the presentation of the sustainable Smart City depends on the interviews of the relevant actors, which is always the case in narrative interviews. For the selection of the case study, ranking systems are used, some of which only consider large cities (over 500,000 inhabitants). Knowledge distribution and dissemination is easier and quicker in large cities than in small ones, so that a transfer of process-related applications to smaller cities may require adaptations and/or supporting structures [62] (pp. 336–337). However, there are also smaller cities, each of which is well networked, for example with surrounding universities, so that it must be examined in each individual case whether the respective city or municipality fulfils the prerequisites for transferring these processes of knowledge expansion and network formation, or whether they need to be adapted in each case. The same applies to very large and rapidly growing cities [62] (p. 337).

The hitherto known applications of innovation biographies proved to be very successful in terms of their knowledge objectives. According to Butzin, [56] innovation biographies

(so far) have been shown to be applicable to the following innovations: organisations, processes, products, services, social and failed innovations [56] (p. 39). In a comparative study using innovation biographies of renewable energies (hydropower, electricity from biogas, solar power generation, geothermal energy and wind power), scientists found driving forces and inhibiting factors in the development processes and derived findings for political control in this area [52] (p. 493). The compilation of ten innovation biographies in the construction industry has also identified numerous good sectoral findings as well as overarching findings on obstacles and drivers, for example, where and how communication and cooperation are important, at what levels learning processes take place and what role motivation plays [52] (pp. 5–6). However, innovation is not limited to new products, but covers entire change processes. The process towards a sustainable Smart City can also be examined in terms of its iterative-recursive character, its development steps and the influencing actor and network structures.

## 6. Conclusions

Sustainable Smart City research is quite advanced in the theoretical field and on a meta-level, as the literature review above has shown. In addition, there are already good approaches to process-oriented analyses within the sustainable Smart City. The innovation biography is intended to expand this portfolio of methods and to offer a new perspective in urban research. With such a method, which captures the sustainable Smart City as a regional innovation system, it is possible to open and sketch the process towards a sustainable Smart City. In addition, further studies must include rebound effects where the sustainable Smart City might be less beneficial than expected. A sustainable Smart City is based on a complex digital construction of technologies and intelligent systems. The foundation for this process must be understood, controlled and used. The cooperation of users, networks and national and international partnerships as well as the processing and dissemination of learning, the combination of citizens' skills and the knowledge of institutions that influence collective learning are at the centre of the consideration as well as the multidimensional character of a sustainable Smart City. The influence of ICT companies and their inventiveness, acting on Schumperter's "pioneer" must be explored [63] (pp. 5–10), for each city is individual in its allocation of resources, such as finances and personnel, as well as in its given infrastructures and the Schumpeter's ideal of an innovator [64]. "[...] the vision about the city of the future is an essential driver of the smart city discourse [...]" [41] (p. 104).

**Funding:** This research was funded by Ministry of Culture and Science of the State of North Rhine-Westphalia as part of the project "Energy efficiency in the neighbourhood".

**Institutional Review Board Statement:** Not applicable.

**Informed Consent Statement:** Not applicable.

**Data Availability Statement:** Not applicable.

**Acknowledgments:** The author would like to thank Ralf Schüle (Federal Institute for Research on Building, Urban Affairs and Spatial Development (BBSR)), Division: Digital Cities, Risk Prevention and Transportation) and Manfred Fischedick (Scientific Managing Director, Wuppertal Institute for Climate, Environment and Energy) for the valuable hints and discussions.

**Conflicts of Interest:** The author declares no conflict of interest.

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
