# Peer review of "Sustainable Smart City—Opening a Black Box"

_sustainability, doi:10.3390/su13020769_

Round 1

Reviewer 1 Report

As far as I can tell there are three parts to this manuscript.

  1. What is the relation between the smart city concept and the sustainable city? This is addressed through an intermittent literature review.
  2. The assertion that the smart city is an innovation which can be understood via a descriptive method, the innovation biography.
  3. An attempt to suggest how this [#2] might be addressed empirically.

The problems with the manuscript are twofold. First, there is no clear connection between the first part and the second. Next, the exploration of the biography is ineffective.

The first half of the manuscript is an endless review of fragments of the literature. This is extremely hard to follow, not least as it seems to touch briefly on all kinds of authors with disparate views.

The second half is of greater potential interest—applying the innovation biography to urban settings [Smart City=innovation]. However, it is not at all clear if the author has the material to accomplish the task and certainly the brief discussion would not suggest how another researcher would proceed to validate the method. Similarly, the attempt at an empirical example is thin and not remotely successful [comparing vast metropolitan regions and comparatively tiny towns like Reykjavik is pointless].

The text states that ‘The innovation biography should show whether these cities are [2120] designated as sustainable cities because or although they use smart solutions’ but ‘should’ is not the same as ‘succeeded’; the sentence is also typical of the complexity of the language used.

All in all, this manuscript is ineffective because the author couples too much ambition with a written style which is never under control.  Even at the close of the paper, the author is trying to tell us what is going on, but remaining at a level of ambiguity—“various formats and methods are available to answer the question of what the tools of urban [2177] design in the 21st century are’.

To be successful, this version—which has not improved a great deal from v1—needs to be torn apart and re-assembled as two totally different papers. The first should directly explore the relation between smart and sustainable cities. This needs to be a coherent argument developed by the author and not a continual effort to tell us what other researchers have said. Bibliometric methods might be the way forward here. I’m enclosing an example which took me 15 minutes and which shows the extent to which smart cities, sustainable cities, smart sustainable cities and sustainable urban development all occupy distinct intellectual spaces (1000 papers published in WoS over past 5 years, ‘smart’ and ‘sustainable’, refined by ‘city’).

The second should explore the innovation biography as it might be applied to a city. Just that. How would it work? Not ‘is it sustainable’ or some other question. Just, how would it be operationalized?  

Author Response

Reviewer 1:

As far as I can tell there are three parts to this manuscript.

  1. What is the relation between the smart city concept and the sustainable city? This is addressed through an intermittent literature review.
  2. The assertion that the smart city is an innovation which can be understood via a descriptive method, the innovation biography.
  3. An attempt to suggest how this [#2] might be addressed empirically.

The problems with the manuscript are twofold. First, there is no clear connection between the first part and the second. Next, the exploration of the biography is ineffective.

Answer: Thank you for your assessment. Of course, I am not sure whether the innovation biography method will work, but I still think this method is suitable because it can represent a dynamic process underlying a Smart City. I hope that this method will be a step forward in the Smart City research to understand how the processes are designed.

Many thanks for the valuable hint to connect the two parts better. I have strengthened the connection between the two parts in some places: in the abstract in lines 10 to 13 and 18 to 19, in the introduction from line 53 to 73, in Chapter 2. in lines 164 to 169, in Chapter 3.1. in line 232

The first half of the manuscript is an endless review of fragments of the literature. This is extremely hard to follow, not least as it seems to touch briefly on all kinds of authors with disparate views.

Answer: Thank you very much for your advice. You are absolutely right, of course, that the presentation in this article is too brief for a complete coverage. However, this would also go beyond the scope of the article. At this point it serves to classify the complexity and versatility in the field of the Smart City and to derive the research gap that is relevant to me: The process analysis, which I expect to show how a Smart City is created.

Thanks for the hint, in this brevity it is no longer a literature analysis, so I have adapted it in line 75 of the headline and called it overview.

The second half is of greater potential interest—applying the innovation biography to urban settings [Smart City=innovation]. However, it is not at all clear if the author has the material to accomplish the task and certainly the brief discussion would not suggest how another researcher would proceed to validate the method. Similarly, the attempt at an empirical example is thin and not remotely successful [comparing vast metropolitan regions and comparatively tiny towns like Reykjavik is pointless].

Answer: Thank you very much for the important note. I have made it clear that this exploratory method naturally has weaknesses with regard to the usual quality criteria. I have added this in the first paragraph of Chapter 5 discussion.

The text states that ‘The innovation biography should show whether these cities are [2120] designated as sustainable cities because or although they use smart solutions’ but ‘should’ is not the same as ‘succeeded’; the sentence is also typical of the complexity of the language used.

Answer: Thank you very much for this information. I have rewritten the sentence.

All in all, this manuscript is ineffective because the author couples too much ambition with a written style which is never under control.  Even at the close of the paper, the author is trying to tell us what is going on, but remaining at a level of ambiguity—“various formats and methods are available to answer the question of what the tools of urban [2177] design in the 21st century are’.

Answer: Thank you for pointing that out, I see your point. I have removed the sentence and had a proofreading done.

To be successful, this version—which has not improved a great deal from v1—needs to be torn apart and re-assembled as two totally different papers. The first should directly explore the relation between smart and sustainable cities. This needs to be a coherent argument developed by the author and not a continual effort to tell us what other researchers have said. Bibliometric methods might be the way forward here. I’m enclosing an example which took me 15 minutes and which shows the extent to which smart cities, sustainable cities, smart sustainable cities and sustainable urban development all occupy distinct intellectual spaces (1000 papers published in WoS over past 5 years, ‘smart’ and ‘sustainable’, refined by ‘city’).

Answer: You are of course absolutely right that only a summary of the numerous literature on Smart City can be given in the short version. On your advice, I have adapted this first part once again and made it clear that it only serves the purpose of classification here, but for me it served (of course to a wider extent) to determine the research gap.

The second should explore the innovation biography as it might be applied to a city. Just that. How would it work? Not ‘is it sustainable’ or some other question. Just, how would it be operationalized?

Answer: Thank you for the good advice, I have strengthened the second part. Nevertheless, the sustainability perspective on the Smart City is very relevant and should not disappear completely. I have tried to illustrate this with examples.

Thank you very much for your detailed and good review of my article. You have helped me a lot to improve it. Also many thanks for the attached graphic.

Submission Date

27 October 2020

Date of this review

30 Oct 2020 20:34:32

Reviewer 2 Report

The article in this version is significantly improved. I believe that after a few more improvements it will be suitable for publication in Sustainability.
I present my comments below.
1. Abstract. The issue of the Biography of Innovation should probably be more exposed. Reading the article, I get the impression that this is the clue of the matter. I can see the author's hesitations on this issue, which unfortunately causes some surprises.
2. Line 261. I cannot agree that only city authorities suffer the risk. In my country, planning firms that make a significant mistake are not invited to cooperate in the future. The flow of information between officials in different cities is very helpful here.
In summary, other actors also suffer the risk.
There is also a third actor - the inhabitants. They also indirectly bear the risk.
3. The discussion should be 2 paragraphs longer. The discussion should refer not only to the idea of ​​Smart City, but also the method of the Biography of Innovation. I would like to know what the author has managed to convey in this matter, in relation to the scientific achievements so far. Are the assumptions presented by other authors correct (or not)?
4. I think it would be good to refer to the idea of ​​diffusion of innovation in the article. This idea has already been applied in social sciences and it would be worthwhile to find differences and similarities. I was very intrigued by the flexibility of statements that an innovator is a person who introduces an innovation or a person who promotes it at subsequent stages. The idea of ​​diffusion of innovation orders this and therefore probably provides an answer to the questions to which the concept of the Biography of Innovation vaguely answers.

Thank you. Idea of the paper is very well. 

Author Response

Reviewer 2:

Answer: Thank you very much for reviewing my article again and for your valuable help

The article in this version is significantly improved. I believe that after a few more improvements it will be suitable for publication in Sustainability.
I present my comments below.

  1. Abstract. The issue of the Biography of Innovation should probably be more exposed. Reading the article, I get the impression that this is the clue of the matter. I can see the author's hesitations on this issue, which unfortunately causes some surprises.

Answer: Many thanks! I have added an introductory sentence to the abstract which complements the idea behind the innovation biography. (lines 10-13).

  1. Line 261. I cannot agree that only city authorities suffer the risk. In my country, planning firms that make a significant mistake are not invited to cooperate in the future. The flow of information between officials in different cities is very helpful here.
    In summary, other actors also suffer the risk.
    There is also a third actor - the inhabitants. They also indirectly bear the risk.

Answer: Thank you very much for your detailed review. You are right, of course. The city planners and, of course, the citizens of a city are also affected. The criticism was directed more at the large information and communication companies. Here cities run the risk of receiving technology and services that they may not need. And yes, communication is certainly a great help here. I have added the inhabitants in line 48.

  1. The discussion should be 2 paragraphs longer. The discussion should refer not only to the idea of ​​Smart City, but also the method of the Biography of Innovation. I would like to know what the author has managed to convey in this matter, in relation to the scientific achievements so far. Are the assumptions presented by other authors correct (or not)?

Answer: Thank you very much for this great proposal. I have supplemented the scientific achievements in the discussion in chapter 5 in the last two paragraphs.

  1. I think it would be good to refer to the idea of ​​diffusion of innovation in the article. This idea has already been applied in social sciences and it would be worthwhile to find differences and similarities. I was very intrigued by the flexibility of statements that an innovator is a person who introduces an innovation or a person who promotes it at subsequent stages. The idea of ​​diffusion of innovation orders this and therefore probably provides an answer to the questions to which the concept of the Biography of Innovation vaguely answers.

Answer. Thank you very much. First of all I had to think for a moment about your reference and read up a little on Rogers' diffusion theory and I also think that this could be a very good addition to the biographies. I have only mentioned it here under the first section in 3.2, but I think that for the implementation of the biography I can consider again to complement this approach. Many thanks for this great tip.

Thank you. Idea of the paper is very well. 

Many thanks for being such a great help!

Reviewer 3 Report

The paper concerns the topical research problem of the creation and development of the structure of 'Smart Cities' and 'Smart and Sustainable Cities'. Among others, it shows the potential of a new and good tool intended for the analysis of the structure of these towns. The paper is worth publishing, but the manuscript structure and content need to be corrected. In particular, I suggest considering the following issues:

  1. In the literature, there are separate concepts of the 'smart city', the 'sustainable city' and the 'smart and sustainable city' (the Author also uses the following terms: 'smart sustainable city', 'smart sustainable cities' and 'sustainable urban design'). The concepts are investigated also in the context of their integration. It is not 100% clear which concept the Author's analysis refers to. I suggest that this should be specified clearly in the text. The title indicates that the analysis is focused on the 'smart city', but from the text as a whole it may rather be concluded that the analysis concerns the 'smart and sustainable city' or the 'Smart City' evolution towards the 'Sustainable City'. For example, the second chapter is devoted entirely to 'smart sustainable city', despite the announcement in the introduction that it will concern 'smart city'.I suggest that the Author should decide which concept the paper is devoted to. Relevant terms should then be used carefully and consistently in the titles of individual sections and in the entire text. Maybe the title of the paper should also be modified?
  2. The paper fails to determine precisely the scope of the analysis of the research problem. The ‘Smart City Black Box’ concept appears in the manuscript. But it is not absolutely clear what the Author means – is he/she concerned with its part only or with its entirety? Does the analysis concern a certain state or changes now in progress? I propose to expand on the author's understanding of the term 'Black Box', especially in the context of the 'Smart City' analysis.
  3. Nobody knows how section 4 contributes to the achievement of the adopted research goal. Is this part of the paper necessary? If yes it should be explained more detailed in the text. I propose to clearly describe in steps and carefully analyze the proposed path of solving the author's research problem and specify the reason for developing chapter 4 in more detail.
  4. As stated by the Author in the Introduction section, the core of the paper is to present a new method that considers Smart City as a regional innovation system. However, the method should be described in greater detail to enhance the research dimension of the paper (subsection 3.2 should be expanded). Maybe it would be appropriate to include a table showing subsequent steps of the proposed method realization and describing, for each individual step, the most important conditions that must be satisfied for the method to be effective for the analysis of the Smart City concept. I also suggest that the method should be additionally compared with other competitive tools. The information resulting from the comparison should be presented synthetically, e.g. using tables comparing the parameters of the methods, such as the scope of the analysis, functionality, etc. Of course, these are just suggestions. The problem can be shown in other ways chosen by the author.
  5. The text also contains some unfortunate statements. For example, lines 8-9 might give the impression that from the outside it is impossible to notice any effects of the digitalization of cities. Lines 27-29 might suggest that carrying out experiments produces no answers, which raises doubts as to whether there is any sense in conducting them. Probably the author of the source publication used these statements in a specific context or showed one of many possible scenarios. It follows from lines 36-38 that the Smart City concept may be either a marketing instrument or an instrument of political control. Are there no other options? Lines 38-40 sound as if there were no studies on the Smart City creation or on the concept relations with Sustainable Urban Development. I suggest a more careful choice of words, as the reader can be misled.
  6. The research problem is not specified in the paper precisely enough. One main aim is indicated (“the aim of the article is to present a method that considers the Smart City as a regional innovation system”). However, there are also many subplots which are not treated consistently throughout the text or in the Conclusions section. I suggest that the research problem(s) be listed in the Introduction section. Then they should be taken into account in subsequent parts of the paper so as not to leave any doubts on part of the reader. I also suggest clearly defining the steps in which the research problem is being solved. It would also be beneficial to show these steps with a diagram.
  7. Although I am not a native speaker, I have noticed some minor linguistic errors.

Author Response

Reviewer 3:

Answer: Thank you very much for your detailed review of my article and the many good tips.

The paper concerns the topical research problem of the creation and development of the structure of 'Smart Cities' and 'Smart and Sustainable Cities'. Among others, it shows the potential of a new and good tool intended for the analysis of the structure of these towns. The paper is worth publishing, but the manuscript structure and content need to be corrected. In particular, I suggest considering the following issues:

  1. In the literature, there are separate concepts of the 'smart city', the 'sustainable city' and the 'smart and sustainable city' (the Author also uses the following terms: 'smart sustainable city', 'smart sustainable cities' and 'sustainable urban design'). The concepts are investigated also in the context of their integration. It is not 100% clear which concept the Author's analysis refers to. I suggest that this should be specified clearly in the text. The title indicates that the analysis is focused on the 'smart city', but from the text as a whole it may rather be concluded that the analysis concerns the 'smart and sustainable city' or the 'Smart City' evolution towards the 'Sustainable City'. For example, the second chapter is devoted entirely to 'smart sustainable city', despite the announcement in the introduction that it will concern 'smart city'.I suggest that the Author should decide which concept the paper is devoted to. Relevant terms should then be used carefully and consistently in the titles of individual sections and in the entire text. Maybe the title of the paper should also be modified?

Answer: Thank you very much. You are absolutely right. I have adjusted the title accordingly, as well as in the abstract and in the introduction in the last paragraph. In the next chapter on the overview of definitions in the field of Smart City, I have also taken up the reference to sustainable Smart City in the last paragraph. Here I made it clear once again that my definition of a Smart City is precisely that which uses its technological solutions for more sustainable urban development.

  1. The paper fails to determine precisely the scope of the analysis of the research problem. The ‘Smart City Black Box’ concept appears in the manuscript. But it is not absolutely clear what the Author means – is he/she concerned with its part only or with its entirety? Does the analysis concern a certain state or changes now in progress? I propose to expand on the author's understanding of the term 'Black Box', especially in the context of the 'Smart City' analysis.

Answer:  Thank you very much for this valuable advice. I have made it clear again directly in the method under 3.1 what is meant by opening the Smart City black box. I also added it again in the first sentence of the abstract.

In the method section under chapter 3, I have reiterated in the last paragraph what biography can do in smart city research - generate a dedicated view of the process towards a smart city.

  1. Nobody knows how section 4 contributes to the achievement of the adopted research goal. Is this part of the paper necessary? If yes it should be explained more detailed in the text. I propose to clearly describe in steps and carefully analyze the proposed path of solving the author's research problem and specify the reason for developing chapter 4 in more detail.

Answer: Thank you very much for this advice.

I have rewritten the first paragraph of the fourth chapter and strengthened the reference to the rest of the article.

  1. As stated by the Author in the Introduction section, the core of the paper is to present a new method that considers Smart City as a regional innovation system. However, the method should be described in greater detail to enhance the research dimension of the paper (subsection 3.2 should be expanded). Maybe it would be appropriate to include a table showing subsequent steps of the proposed method realization and describing, for each individual step, the most important conditions that must be satisfied for the method to be effective for the analysis of the Smart City concept. I also suggest that the method should be additionally compared with other competitive tools. The information resulting from the comparison should be presented synthetically, e.g. using tables comparing the parameters of the methods, such as the scope of the analysis, functionality, etc. Of course, these are just suggestions. The problem can be shown in other ways chosen by the author.

Answer: Thank you very much for this good advice. I have added a table with dedicated steps of the method and possible challenges in smart city research. This was really helpful.

I plan to present a comparison with competing methods, but in the following article, when I actually present the method in practice. Due to the Corona pandemic, it is very difficult to get interview appointments in the cities, they simply have a lot of other things to do at the moment. But thank you very much for this valuable tip.

  1. The text also contains some unfortunate statements. For example, lines 8-9 might give the impression that from the outside it is impossible to notice any effects of the digitalization of cities. Lines 27-29 might suggest that carrying out experiments produces no answers, which raises doubts as to whether there is any sense in conducting them. Probably the author of the source publication used these statements in a specific context or showed one of many possible scenarios. It follows from lines 36-38 that the Smart City concept may be either a marketing instrument or an instrument of political control. Are there no other options? Lines 38-40 sound as if there were no studies on the Smart City creation or on the concept relations with Sustainable Urban Development. I suggest a more careful choice of words, as the reader can be misled.

Answer: Many thanks for your detailed review. I have adapted or removed all their statements. Lines 8-9 I describe in more detail in lines 10-13. In lines 36-38 (in the current version lines 41-42) I have added the following: a vision, a marketing instrument, a political control instrument or something else entirely. I have also adapted the sentence that follows. Many thanks for your detailed review.

  1. The research problem is not specified in the paper precisely enough. One main aim is indicated (“the aim of the article is to present a method that considers the Smart City as a regional innovation system”). However, there are also many subplots which are not treated consistently throughout the text or in the Conclusions section. I suggest that the research problem(s) be listed in the Introduction section. Then they should be taken into account in subsequent parts of the paper so as not to leave any doubts on part of the reader. I also suggest clearly defining the steps in which the research problem is being solved. It would also be beneficial to show these steps with a diagram.

Answer: Thank you very much for this good advice. What have I changed in this respect: Abstract strengthened with regard to the research question, as well as the introduction in the 3rd and 4th paragraphs. I also rewrote the last two paragraphs in chapter 2 and added examples in the first paragraph under 3.1. I have also added lines 270-279 and under discussion in chapter 5 where these innovation biographies have been applied and what results they have produced.

  1. Although I am not a native speaker, I have noticed some minor linguistic errors.

Reviewer 4 Report

Line 16, page 1, under abstract: instead of ‘und’ should be and.

Line 1393, page 4: delete ‘smart’, and should read a city is a smart city.

Line 1466, page 5: remove ‘it’

Line 1657, page 6: remove ‘a’ before further.

Author Response

Reviewer 4

Thank you for reviewing it and pointing out the errors. I have also given it to an external proof reading

Line 16, page 1, under abstract: instead of ‘und’ should be and.

Answer: Thank you very much, I must have overlooked that.

Line 1393, page 4: delete ‘smart’, and should read a city is a smart city.

Answer: Thank you. I have corrected that in line 169

Line 1466, page 5: remove ‘it’

Answer: Thank you very much, I have rearranged the sentence and recognised the error.

Line 1657, page 6: remove ‘a’ before further.

Answer: Thank you. I have corrected that.

Round 2

Reviewer 1 Report

.

Reviewer 2 Report

Dear Author,

The current version of the article is very good. The article is interesting. They recommend it for posting. Only figure 2 needs improvement.
In the title, write which country it is and the contrasts can be slightly improved, so that the visible elements are more readable.

Best regards

Reviewer 3 Report

The paper is worth publishing, but in my opinion the manuscript structure still needs improving. The Author’s main aim is to describe a method that “will highlight which actors are involved, how knowledge is shared among them, what form citizen participation processes take and whether the use of digital and smart services within a Smart City leads to a more sustainable city. Such a process-oriented method should show, among other things, to what extent and when sustainability-relevant motives play a role and which actors and citizens are involved in the process at all”. However, the Author herself says that the method presented in the paper has not been used for this purpose yet – “This question is to be answered by a method that has not yet been applied in research on cities and smart cities: the innovation biography”. The description of the method itself and the possibilities of its application should therefore be the key element of the paper and, as such, adequately highlighted in the text. So far the description of the method is merely a part of the section entitled “Methods employed in Smart City Research and Smart Sustainable City Research and some of their findings”. This is illogical.

The paper still fails to present a clear description of the applied research methods or the research process. What methods did the Author use and/or in what steps did she conduct the analysis of the possibility of utilizing the innovation biography method to realize the new goal?

The title of section 4 (“Case study selection”) is not precise enough – it suggests that the case study was selected to perform the analysis described in the paper. But in fact it is a proposal for case studies to investigate in future analyses. The key issue is the way and not the result, so maybe it should be a part of the described method?